# Thermal Comfort Comparison and Cause Analysis of Low-Temperature High-Humidity Indoor Environments of Rural Houses in Gansu Province, China

Junjie Li [1,2], Xijun Wu [1,*], Sharon K. W. Chow [3], Qiushi Zhuang [1] and Guillaume Habert [2]

1    School of Architecture and Design, Beijing Jiaotong University, Beijing 100044, China;
     lijunjie@bjtu.edu.cn (J.L.); 22140416@bjtu.edu.cn (Q.Z.)
2    Chair of Sustainable Construction, ETH Zürich, 8092 Zurich, Switzerland; habert@ibi.baug.ethz.ch
3    Collaborating Centre for Oxford University and CUHK for Disaster and Medical Humanitarian
     Response (CCOUC), The Chinese University of Hong Kong (CUHK), Hong Kong SAR, China;
     sharon.chow@link.cuhk.edu.hk
*    Correspondence: 21121743@bjtu.edu.cn; Tel.: +86-10-18801001465

**Abstract:** Low temperatures and high humidity often occur in the northern basins and mountainous regions of China. This research reveals a common winter indoor environment in this rural areas characterized by low-temperature and high-humidity indoor thermal conditions. Improving this environment directly with equipment would inevitably result in significant energy consumption. Therefore, comprehending the thermal performance mechanisms of different structural building materials is of vital importance as it provides crucial baseline values for environmental improvement. This study conducted a survey utilizing user questionnaires, resulting in the collection of 214 valid responses. Additionally, a local experiment regarding thermal comfort was conducted. Simultaneously, this study monitored the indoor physical environments of these houses (a sample of 10 rooms was taken from earth houses and 12 rooms from brick houses). Parameters measured on site included air temperature, relative humidity, light illumination, and $CO_2$. The results showed that the humidity inside the earth houses is more stable and regression models can be developed between thermal sensations and temperature for long-term residents. The residents of these earth houses are more sensitive to temperature step. In contrast, the residents of brick houses, experiencing greater environmental variability, demonstrated lower sensitivity and greater adaptablity to temperature changes. In addition, heating from bottom to top is more comfortable and healthier for the residents of brick houses in Gansu. Moreover, it is more favorable for the inhabitants' livelihood to regulate the temperature steps to a maximum of 4 °C. This study provides valuable reference information for the future design of houses in low-temperature and high-humidity environments.

**Keywords:** thermal environment; earth house; brick house; cold climate zone

## 1. Introduction

### 1.1. Research Background

   Buildings are responsible for approximately 39% of global $CO_2$ emissions. A breakdown of this figure reveals that 28% of global $CO_2$ emissions can be attributed to building operations—commonly referred to as operational carbon; while the remaining 11% came from building materials and construction—which was known as embodied carbon [1]. According to EIA (Electronic Industries Alliance) annual report data, China was the world's largest energy consumer, and its energy demand was expected to continue to grow [2]. The construction and operation of China's buildings produced 42% of the country's total greenhouse gas emissions [3]. In September 2020, China proposed a 2030 "carbon peak" and "carbon neutrality" by 2060. This double carbon target has been a goal of the construction industry ever since. According to the 2021 China Statistical Yearbook [4], the

rural population accounted for 36% of the total population of the country. In 2016, the heating area of rural buildings in northern China was about 6.5 billion m$^2$, accounting for 31.6% of the total heating area of northern buildings [5]. In contrast to urban areas, the advancement of energy efficiency in rural buildings has comparatively fallen behind in terms of incorporating energy-saving design principles and implementing energy-conserving infrastructure. A large number of new housing units in rural areas were built with less professional guidance and sophisticated construction techniques, resulting in buildings with poor indoor thermal environment.

In another aspect, to ensure a healthy indoor thermal environment, rural dwellers must use a significant amount of energy to maintain a stable indoor environment. The most widely used heating devices in rural region of Gansu are coal stoves and Chinese Kang (an archaic, integrated residential mechanism designed to accommodate culinary endeavors, nocturnal slumber, domestic warming, and ventilation for temperature regulation [6–8]); both of which can cause indoor air pollution. According to a survey, long-term living in this environment can lead to an increased probability of respiratory diseases [9]. Therefore, enhancing the ability of houses to withstand the harsh environment will effectively reduce energy use for heating and facilitate the creation of a healthy home. At present, there was a wealth of research on the thermal environment of typical buildings in various climatic zones in China [10–12]. However, current research lacks an exploration of the differential sensitivity to temperature steps among long-term rural inhabitants across different houses. Thus, this study discussed the environmental effects of two types of houses in southern Gansu, temperature steps on the thermal sensation vote, and the thermal comfort vote outcomes of their residents. This study elucidated the disparities in performance between earth houses and brick houses under specific climatic conditions, characterized by low temperatures and high humidity. It unraveled the potential further exacerbation or alleviation of such conditions by these two types of dwellings, while also exploring their impact on human comfort. In light of the prevailing global resource scarcity and the daunting reality of population aging, home-based elderly care remains the prevailing trend, particularly in China. The government strongly advocates for both home living and elderly care, underscoring the imperative of investigating the influence of architectural environments on individuals and the significant implications for future architectural renovations and designs.

*1.2. Objective of This Study*

In China, rural areas accounted for 70% of the country's total land area and are inhabited by 43% of its population. However, there exists a significant disparity in the distribution of rural comprehensive development levels. With the continuous economic development, rural areas have become a focal point for construction. Nevertheless, if the same construction methods as those used in urban areas are applied, it may lead to higher carbon emissions and energy consumption, posing a direct threat to environmental sustainability. Therefore, this study focuses on a typical rural area in western China and conducts detailed surveys and on-site investigations, primarily addressing the following two research questions:

(1) In a low-temperature, high-humidity environment, what is the underlying mechanism behind the real physical performance of low-cost indigenous materials?
(2) How can houses be made more conducive to comfort and health while keeping economic costs limited?

This study investigates the thermal performance of two prevalent rural building structures and the distinct conditions experienced by the inhabitants. It provides evidence of the subtle differences in living environments that can have a lasting impact on individuals over time. Moreover, it offers some temperature setting references for the design of residential environments in low-temperature, high-humidity conditions.

## 2. Methodology

### 2.1. Study of the Current Situation

The region of Gansu, situated within China, can be classified as a cold region based on China's thermal zoning system, while falling within the temperate zone as per the Köppen climate classification. This study selected a developing village in Tianshui, Gansu Province, China (see Figure 1). Datan Village was monitored over a period of winter. The coldest month is January, with an average temperature of −6.0 °C, and the south of Gansu province has substantial precipitation and is considered a wet zone [13]. The traditional house envelope in the Tianshui city of Gansu is mainly of earth construction, therefore, during the winter season, due to the insufficient heating capacity, residences often exhibit a phenomenon of low temperature but high humidity, significantly affecting human comfort. The primary reason of high humidity is the winter snowfall in this region, which leads to moisture absorption by the earth-built structures [14]. Since 2012, some researchers have focused on the strong regional characteristics of the area's traditional earth constructions, summarizing their construction techniques and structural features [15,16]. Z. N An [17] classified the sink ability class of local clay in the Tianshui city. Other researchers have analyzed survey results, finding that the adobe walls of earth construction offer good abilities to buffer humidity, and favorable thermostability [18]. However, the monitoring of one room did not offer an objective result.

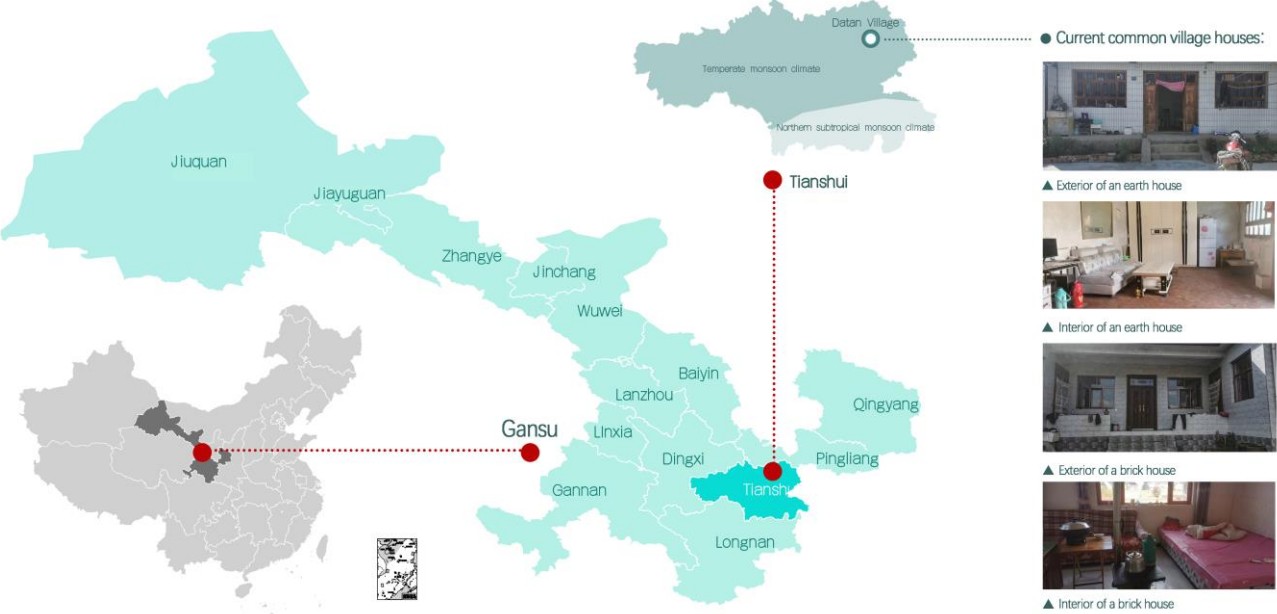

**Figure 1.** Location map of the selected villages.

The local clay is easily available, so most of the buildings in the village more than 20 years old are earth houses. Over the past two decades, many brick houses in the village were demolished and rebuilt on the original courtyard. The dimensions of the living rooms in the local residences extended from 25 to 40 m², with a notable prevalence of rooms oriented towards the south, north, and east, while the number of west-facing rooms remains relatively sparse. A group of recently constructed dwellings predominantly consists of two stories, encompassing a land area of approximately 100 m².

The Datan village is a traditional minority nationality village with 243 villagers. The buildings in the village can be divided by main envelope material: earth or brick. The largest proportion of the villages' dwellings are constructed with earth, accounting for approximately 60% of the village. Furthermore, the majority of the inhabitants live in this type of housing on a permanent basis. Earth houses are traditional homes built with soil and wood as the outer envelope. The walls of these buildings are made of unsintered soil

mixed with sandy clay at a certain proportion [19]. This type of wall preserves heat and features good moisture buffering abilities [20,21]. Most houses in the area are one-story bungalows, with tiles and other decorations on the walls. In addition, there are a large number of brick buildings in the village, some of which are new projects built by villagers self-renovating their own homes (see Figure 1). Others are two-story houses recently built by government builders under the guidance of public policy. The typical layout is depicted in Figure 2, where the courtyards are arranged in a relatively orderly manner, with minimal deviation in the angles of the houses.

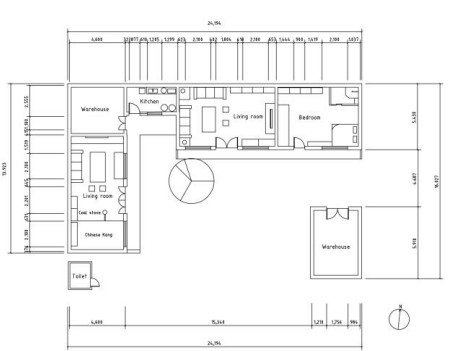 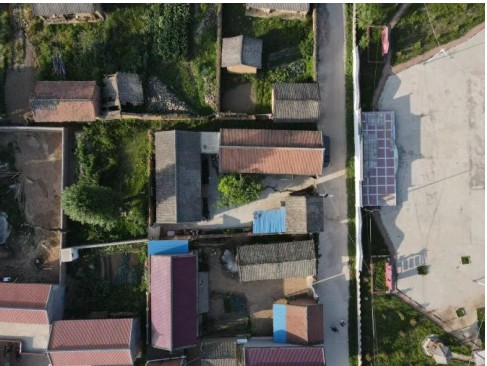

**Figure 2.** Floor plans and aerial views of exemplary buildings.

### 2.2. Questionnaires

This study was divided into two phases. The first was one questionnaire on thermal comfort issues and one questionnaire on the current living conditions of the local population. The second phase involved selecting local representative houses for onsite monitoring. The former employed a census-style questionnaire, while the latter employed a questionnaire tailored to match the actual testing conditions. Although the testing season remains consistent, the questions themselves differ.

During the research period, 243 villagers participated in the completion of questionnaires. Approximately 53.27% of the population consists of males, while females account for approximately 46.95%. Among them, around 40% of individuals are aged over 60. It is worth noting that all respondents in the survey are free from significant ailments and possess the ability to live independently. A total of 214 valid questionnaires were returned, for a villager participation coverage rate of 88% (a level sufficient for sociological census research). The main content of the questionnaire included five sections: (1) basic information about the residents; (2) construction envelope; (3) conditions of heating, ventilation, and air conditioning systems; (4) residents' thermal sensation vote (TSV); and (5) residents' health and hygiene conditions. Under the guidance of the local village committee, the research team completed the questionnaire records within a week's time. The onsite research is shown in Figure 3.

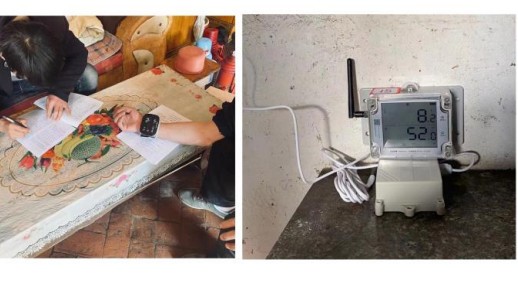 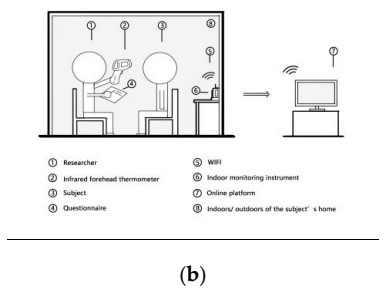

(**a**)                                                                 (**b**)

**Figure 3.** Local resident interview questionnaires and instruments. (**a**) Questionnaire interview and specific instrument. (**b**) Field work diagram.

At the same time as the sociological questionnaire was distributed, the research team identified the physical parameters of the actual environment. Before the questionnaire research began, temperature and humidity measurement instruments were placed to ensure the accuracy of the instrument data. Subsequently, the questionnaire interviews began.

In February 2023, the team sampled 50 individuals in Datan village to conduct a questionnaire survey on thermal comfort, which included an evaluation of the existing environment and the effects of temperature steps on various parts of human body. The outdoor temperatures during this study were all between −1 °C to 10 °C, the average outdoor relative humidity was 55%, the BMI (Body Mass Index) were all between 19–25, and the clothing worn was basically the same (cotton jackets, jumpers and cotton trousers). The specific process was as follows and the operation is shown in Figure 3b:

(1) In the respondent's home, turn on the temperature thermometer and place it at a height of approximately 1.2 m (out of direct sunlight).
(2) After, the respondent rests for 5–10 min. Then, measure the body temperature of respondent, the room temperature and humidity.
(3) Questionnaire interview about indoor environment.
(4) After remaining indoors for 2–5 min, respondents were asked to move outdoors for 10 min without changing their clothing.
(5) Questionnaire interview about outdoor environment.

The questionnaire included a basic physical description of the population and a general assessment of the indoor environment (in Appendix A). The indoor and outdoor sections of the questionnaire included the entire thermal sensation vote (TSV) of temperature steps, as well as the thermal sensation vote (TSV) of various body parts and the thermal comfort vote (TCV).

Following the completion of this survey, the research team utilized SPSS 26 for preliminary questionnaire analysis. Subsequently, the obtained results were imported into Origin 2021 for the final processing stage.

*2.3. On-Site Monitoring Method*

Ubibot indoor monitoring instruments were selected for this study and carbon dioxide sensor was connected for joint monitoring. The instrument can simultaneously monitor temperature, humidity, indoor light conditions, and indoor carbon dioxide concentration. The specific accuracy range can be seen in Table 1. This instrument can be connected to the network by turning on the device, where it records those data to the online platform every five minutes. This instrument sends reminders when the network is disconnected or data are abnormal. Specific instrument parameters are shown in Table 1. Instrument parameters are in accordance with the relevant standards [22,23].

**Table 1.** Parameters of the Monitoring Instruments.

| Equipment Model | Measurement Data and Units | Measurement Range | Measurement Accuracy | t/min |
|---|---|---|---|---|
| Ubibot GS1 AL4G1RS | Temperature (°C) | −20 °C–60 °C | ±0.3 °C | 5 |
| | Relative Humidity (%) | 10–90% | ±3%RH | 5 |
| | Light (lux) | 0.01 to 83K lux | 2% | 5 |
| RS485 $CO_2$ sensor | Carbon dioxide concentration (ppm) (0–50 °C, 0–95% RH) | 0~10,000 ppm | ±(30 ppm + 3%F.S) | 5 |

According to the interviews, local residents tended to spend the winter in traditional earth houses and use coal stoves to heat their homes, some of which are connected to traditional Chinese Kang. The kitchen space is often used for this custom throughout the year, except in winter. However, due to the low winter temperatures, the lives of the entire family were concentrated in the living room, with a coal stove to ensure the

necessary thermal comfort, as well as a fire for cooking and other activities. Toilets are built outside due to the lack of sewerage. Even in the new two-story brick houses, there is only a washroom inside. Interviews revealed that repeated travel between the indoor and outdoor spaces during the winter months was a major problem. And the temperature steps caused discomfort and potential health risks. To investigate the performance of different housing types in low-temperature and high-humidity environments, and the relationship between housing and thermal comfort, the team carried out onsite monitoring twice in winter. The ideas are shown in Figure 4. The area is generally cool and has a high level of radiation in summer, so that some residents still use Chinese Kang for heating in the summer. Consequently, a third survey was conducted for the rooms previously monitored during winter. In the course of this procedure, this study has collectively monitored the specific allocation of primary utilitarian spaces in a total of 22 houses. The detailed arrangement is as follows:

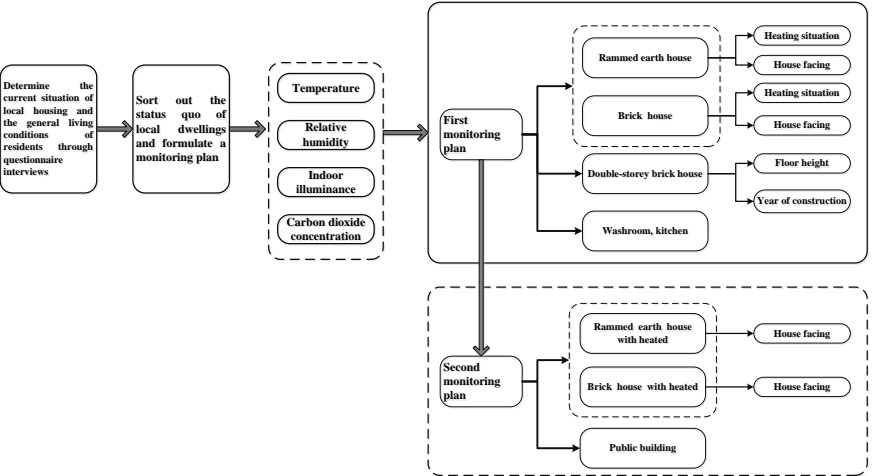

**Figure 4.** Monitoring program development process.

The initial monitoring took place over a span of seven consecutive days, from 19 to 26 January 2022. Two typical building types in the village were selected for monitoring, with the control groups being south-facing vs. non-south-facing, with heating vs. without heating, and one story vs. two stories. Houses with different monitoring organizations and representative rooms were sampled for instrument placement, under the guidance of the villagers (see Table 2).

The second monitoring was carried out over a period of seven days from the 12th to 18th of February in the year 2022. Additional sampling efforts were undertaken specifically targeting earth and brick houses with heating. Three houses with different envelopes and orientations were selected as the focal points of our analysis (see Table 2).

The third phase of monitoring was conducted over a seven-day period from 22nd July to 29th of July in the year 2022. This particular monitoring endeavor targeted specific rooms that were previously monitored in the initial monitoring.

Due to the fact that all local constructions are self-built, there is no specific provision for insulation and moisture prevention. However, the general approach is similar, with the window-to-wall ratio of the main-facing rooms kept within the range of 0.2 to 0.3. Currently, the typical enclosure structure for earth houses consists of a 20 mm layer of thatch, followed by 360 mm of adobe bricks, and another 20 mm layer of thatch. For brick houses, the common practice for the enclosure structure involves a 20 mm layer of cement mortar, followed by 240 mm of fired bricks, and another 20 mm layer of cement mortar. During the monitoring and research process, indoor ventilation was restricted, and mechanical systems were not employed, with only doors being opened for users' life regularly. The instruments were uniformly placed at a height of 1.2 m from the ground, with no obstructions or direct sunlight.

**Table 2.** Test Instrument Placement and Numbers.

| Phase | Mapping Location | Sample Photo | Instrument Placement | Number of People Using the Room | Room Size/m² | Duration of Use/Years | House No. |
|---|---|---|---|---|---|---|---|
| Phase 1 | Earth house |  | South-facing room with heating | 6 | 16 | 23 | E1 |
| | | | Non-south-facing room with heating | 4 | 14 | 23 | E2 |
| | | | South-facing room without heating | 3 | 11 | 30 | E3 |
| | | | Non-south-facing room without heating | 7 | 15 | 40 | E4 |
| | Brick house | | South-facing room with heating | 10 | 21 | 4 | B1 |
| | | | Non-south-facing room with heating | 4 | 26 | 5 | B2 |
| | | | South-facing room without heating | 4 | 21 | 5 | B3 |
| | | | Non-south-facing room without heating | 4 | 18 | 5 | B4 |
| | Two-story brick house |  | 1st floor south-facing room with heating | 6 | 24 | 2 | D1 |
| | | | 2nd floor south-facing room without heating | 0 | 15 | 2 | D2 |
| | New two-story house |  | 1st floor south-facing room with heating | 3 | 26 | 3 | D3 |
| | | | 2nd floor south-facing room without heating | 3 | 22 | 3 | D4 |
| | | | Indoor washroom | | 3 | 3 | W1 |
| | Washroom (outdoor toilet) | | Hanging against the wall | | 2 | 23 | W2 |
| | Kitchen | | Kitchen in a earth house | | 5 | 30 | K |
| | | | | | | | O |
| Phase 2 | Earth house |  | South-facing room with heating | 4 | 23 | 10 | EE1 |
| | | | | 6 | 16 | 23 | EE2 |
| | | | | 4 | 20 | 23 | EE3 |
| | | | Non-south-facing room with heating | 3 | 18 | 20 | EE4 |
| | | | | 6 | 26 | 30 | EE5 |
| | | | | 1 | 30 | 32 | EE6 |
| | Brick house |  | South-facing room with heating | 3 | 20 | 6 | BB1 |
| | | | | 7 | 28 | 6 | BB2 |
| | | | | 4 | 24 | 10 | BB3 |
| | | | Non-south-facing room with heating | 2 | 23 | 14 | BB4 |
| | | | | 5 | 12 | 7 | BB5 |
| | | | | 6 | 26 | 3 | BB6 |
| | Outdoor | | | | | | OO |

Moreover, the heating method using Chinese Kang is challenging to control. During the research period, the temperature was relatively low and coincided with Chinese holidays. To ensure stable residential heating, each household generally adopted continuous heating throughout the day.

For the analysis of the measured data, a preliminary screening was conducted using EXCEL, selecting reliable data from the same time period. The selected data was then imported into SPSS 26 and Origin 2021 for further processing. SPSS 26 was utilized to examine the correlation between temperature and humidity, and regression analysis was performed with respect to temperature. Subsequently, the main findings of this study were

presented through image processing in Origin 2021. The locations of the selected rooms are as shown in Figure 5.

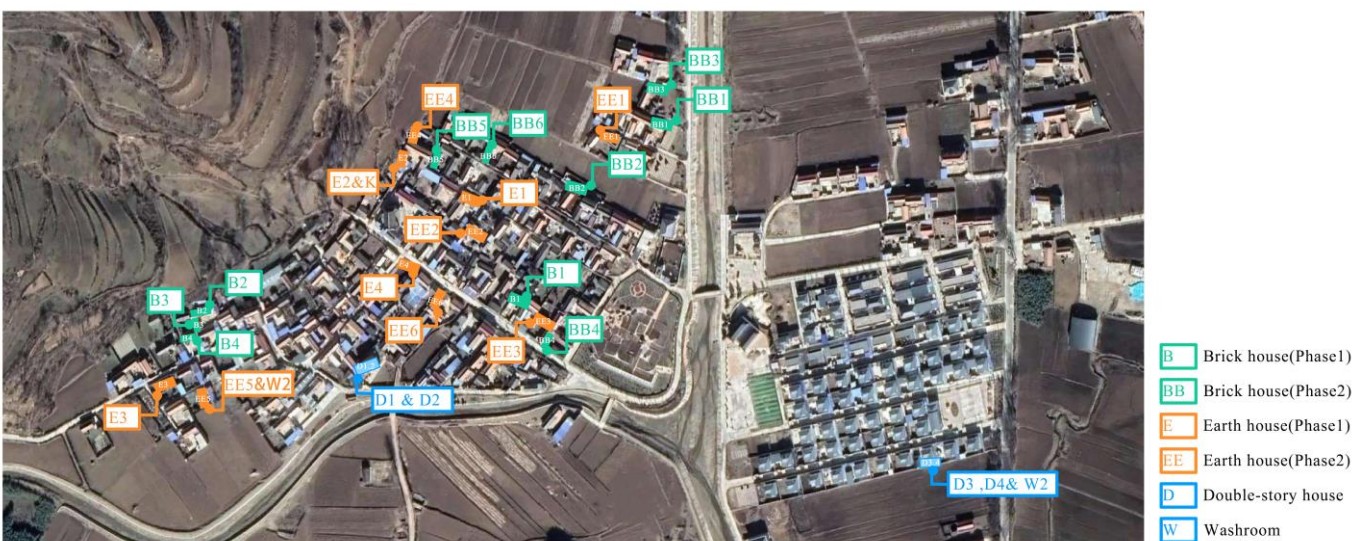

**Figure 5.** Floor plan of selected building test points.

## 3. Summary of Research Findings and Characteristics

### 3.1. Results of the Indoor Thermal Environment Study

After comparing the current state of affairs with the existing regulations concerning public health in Table 3, it can be found that International Health Regulations [24] do not specifically address residential buildings, but rather focus on regulations pertaining to public structures. The ASHARE Guideline for the Risk Management of Public Health and Safety in Buildings places a greater emphasis on the emergency functionalities and siting of buildings [25]. Neither of the aforementioned regulations are applicable.

The WELL V2 standard provides detailed provisions for promoting healthy architecture [26]. Therefore, this study compares the key provisions that are relevant and adaptable to the current situation in rural China. The results show that the control of air quality has generally been neglected, while the management of factors related to thermal comfort is also lacking. The impact of noise must be duly acknowledged, and there is a notable dearth in terms of the community's conception of development. The buildings in this area are in need of upgrading. Furthermore, the rural areas of Gansu Province encompass a wide expanse and have a dense population [4], presenting ample opportunities for development.

3.1.1. Temperature and Relative Humidity Performance of Earth Houses

This study employs two different criteria to compare the variations in indoor temperature and humidity of the buildings. Firstly, based on the ASHRAE 55.92 [27], the recommended indoor effective temperature during winter should range between 20–23.5 °C, while during summer it is advised to be within 23–26 °C. The relative humidity should be maintained between 30% and 60%. Secondly, in accordance with the Indoor Air Quality Standard (China), temperature fluctuations should be within 16–24 °C, and relative humidity fluctuations should be between 30–60% in winter. And in summer, temperature should be within 22–28 °C, and relative humidity fluctuations should be between 40–80% [28]. Both contrasting ranges have been indicated in the figure. In this regard, it is noteworthy that the humidity requirements during the winter season in China correspond harmoniously with the recommended guidelines stipulated in the ASHARE 55.92. Consequently, this particular aspect has not been replicated within the depicted visual representation.

**Table 3.** The overall condition of the rural areas' house.

| International Health Regulations [24] | ASHARE Guideline 29-2019 [25] Guideline for the Risk Management of Public Health and Safety in Buildings | WELL V2 Standard [26] | | Circumstances of Houses within This Study |
|---|---|---|---|---|
| | | Type | Content | |
| This regulation does not encompass provisions regarding the hygiene conditions of residential buildings, but it places greater emphasis on the requirements for facilities within public buildings. | The requirements for the indoor environment of buildings in this standard are unclear. Specific requirements for building envelopes should be established to regulate energy transfer and minimize air and vapor transportation. Additionally, it is recommended to provide emergency and energy storage systems. | Air | In the Fundamental Air Quality section, it is mentioned that projects must ensure the provision of air quality levels deemed acceptable by public health authorities. In the section dedicated to Implementing Demand-Controlled Ventilation, it is specified that the indoor carbon dioxide concentration must be below 900 ppm in order to fall within the scoring range. | Those houses rely on coal stoves and heated brick beds for warming purposes, resulting in significant air pollution. This heating methodology deviates from the standards of healthy constructions. During the monitoring process, it was observed that the local heated rooms suffer from poor ventilation, leading to elevated levels of carbon dioxide. The $CO_2$ content in all the monitored heated rooms exceeded 900 ppm. This represents an average increase of over 300 ppm compared to the outdoor carbon dioxide concentration. |
| | | Nourishment | This endeavor necessitates the accessibility of an assortment of fruits and vegetables, while emphasizing the importance of transparency in terms of nutritional value. | In the rural areas of China, it is possible to cultivate all kinds of vegetables and other food products for personal consumption, ensuring freshness and promoting health. But according to surveys, local residents have a habit of consuming pickled vegetables. |
| | | Light | Regarding the stipulations applicable to residential premises, it is mandated that the extent of transparent envelope glazing shall not be lesser than 7% of the total floor area. | The local area experiences strong sunlight radiation, and the selected building windows have an average transparency of around 0.3, which generally meets the requirements. |
| | | Movement | As per the provisions of this regulation, it is mandated that both the community and the interior of buildings allocate facilities for physical exercise, further necessitating the engagement of residents in regular physical activities. | Under the impetus of the government, the rural areas of the region have all been equipped with fitness apparatus. But their households lack the requisite infrastructure for fitness facilities. |
| | | Thermal Comfort | In the Thermal Comfort Monitors section, it mandates the installation of a sensor on each floor at a minimum frequency of one per 325 square meters, with a placement distance of no less than 1 m from the primary heat or cold source. | The placement positions of the instruments in this study all adhere to the specified criteria. |
| | | | In the Humidity Control section, It stipulates that the relative humidity in primary occupancy areas should be regulated within the range of 30% to 60%. | Humidity control is rarely implemented in the local area. |
| | | | In the Individual Thermal Control section, it requires the project to provide individual thermal comfort control devices or flexible dressing strategies. | The local area lacks any measures to ensure this aspect. |
| | | Sound | It is required to take into consideration the impact of noise on health, as well as to implement measures for its control and mitigation. | Based on the survey results, the majority of local residents have not implemented soundproofing facilities. Furthermore, the survey reveals that only 54.23% of individuals acknowledge that noise has not affected them personally. Many residents perceive the impact of noise on their well-being. |
| | | Materials | This provision dictates the imperative need for stringent regulation of hazardous substances inherent in building materials. | The local materials predominantly comprise indigenous resources, with minimal utilization of interior decoration. |
| | | Community; Innovations; Water | These three requirements encompass the principles of communal co-creation, innovative design, and the assurance of water quality. | Based on the interviews conducted, it has been ascertained that the community in this area fosters amicable neighborly relations, while the provision of water resources is uniformly managed through municipal engineering projects. |

In the initial monitoring results (see Table 4), the earth houses without heating had a slightly higher temperature, as evidenced by the south-facing E3 being greater than the non-south-facing E4 (see Figure 6). Both featured a relative humidity of above 80% and temperatures fluctuated at 0 °C. The indoor environment of the earth houses without heating showed a distinctly low temperature and high-humidity environment which was unsuitable for winter habitation.

**Table 4.** Descriptive Statistics for Temperature and Humidity in the Earth Houses.

| | | Minimum Value | Maximum Value | Average Value | Standard Deviation | Variance |
|---|---|---|---|---|---|---|
| E1 | Temperature (°C) | 9.287785 | 16.051727 | 12.61588200 | 1.729110480 | 2.990 |
| | Relative Humidity (%) | 37 | 86 | 49.19 | 4.955 | 24.548 |
| E2 | Temperature (°C) | 7.063400 | 14.249256 | 10.18938402 | 1.394559673 | 1.945 |
| | Relative Humidity (%) | 51 | 78 | 60.12 | 3.971 | 15.767 |
| E3 | Temperature (°C) | −0.533684 | 1.808193 | 0.49046708 | 0.530403608 | 0.281 |
| | Relative Humidity (%) | 81 | 91 | 88.15 | 2.490 | 6.199 |
| E4 | Temperature (°C) | −2.029068 | −0.733959 | −1.25271162 | 0.377199255 | 0.142 |
| | Relative Humidity (%) | 73 | 81 | 78.34 | 1.365 | 1.863 |

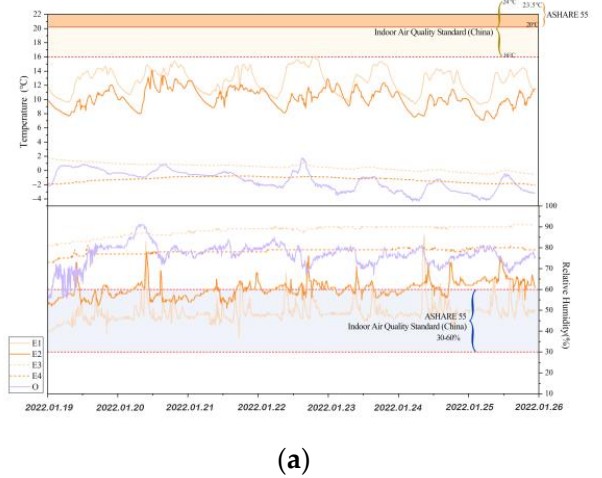

(**a**)

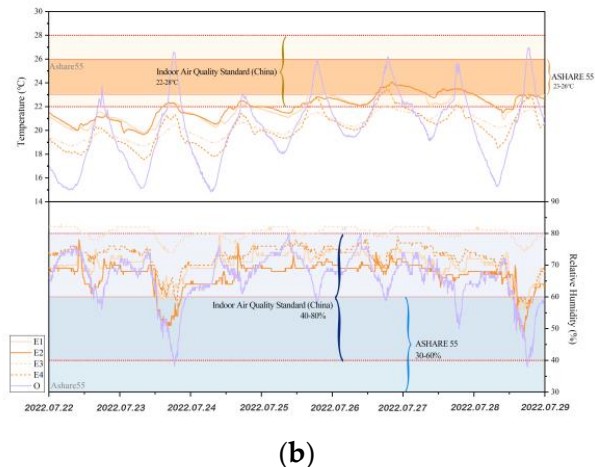

(**b**)

**Figure 6.** Temperature and Relative humidity of earth houses in one week. The recommended temperature range for the ASHARE 55 standard is delineated in orange, while the humidity range is delineated in blue. The temperature range specified by the Chinese Indoor Air Quality standard is delineated in yellow, and the humidity range is delineated in light purple. As the humidity setting for both standards is the same in winter, only one range is delineated. Additionally, the specific range corresponding to each standard is labeled with text). (**a**) Temperature and RH fluctuation in winter. (**b**) Temperature and RH (Relative Humidity) fluctuation in summer.

In contrast, the indoor temperature experienced a considerable rise and the relative humidity exhibited fluctuations within the range of 30% to 60% with heating. This fulfilled the pertinent criteria outlined in the standard for winter heating indoor parameters [28]. The temperature and relative humidity during this phase were in line with the characteristics of the earth house itself: superior moisture buffering abilities [20]. As depicted in Figure 6a, none of the earth houses with heating reached the design values in the design standards for heating temperatures, but the temperature in the south-facing E1 was slightly higher compared to non-south-facing E2, and the relative humidity in the former was also more in line with the design values. This indicates that the indoor environment can be effectively enhanced by increasing the temperature of the room when the interior of an earth house features a low-temperature, high-humidity environment.

In addition, the graphical representation in Figure 6a showed that the temperature and relative humidity of E1 and E2 were very similar to one another, which proves the effect of the different orientations was not significant with heating. The overall temperatures of the unheated rooms were similar to the outdoor temperature, and the temperatures of south-facing rooms were slightly higher. Therefore, the building orientation had less influence on the overall effect of the earth houses, but south-facing rooms still enjoyed certain orientation advantages in Gansu.

In the summer, as shown in Figure 6b, the temperatures in Datan village are low compared to the city and the climate is generally high-humidity. The envelope of the earth houses were continuously damped due to the increased temperature. The entire relative humidity inside the rooms was high, which in turn affects the room temperature. The indoor temperature was lower than the outdoor temperature and slightly below the temperature specified in the standards [28]. However, the relative humidity was within the standard range [28] but higher than the outside relative humidity. In order to enhance comfort levels, the relative humidity can be reduced by increasing ventilation [29]. However, the relative humidity levels during the summer season still remain relatively elevated compared to the recommended standards set forth by ASHRAE.

### 3.1.2. Temperature and Relative Humidity Performance of Brick Houses

The winter monitoring findings for the brick houses are shown in Table 5. According to Figure 7a, there was a large difference in temperature between the south-facing D2 and the non-south-facing B3 and B4 without heating. Evidently, south-facing rooms were significantly warmer than non-south-facing rooms. Specifically, the average temperature the south-facing rooms stood at 4.96 °C, while the non-south-facing rooms was approximately −3.29 °C during a week monitoring. There was little difference in relative humidity. The average relative humidity of the south-facing brick houses was approximately 76%, while the average relative humidity of the non-south-facing brick houses was approximately 70.82%, both showing high relative humidity. The brick houses had poor moisture buffering abilities, and condensation occurred on the internal surfaces of the walls, resulting in higher indoor humidity.

**Table 5.** Descriptive Statistics for Temperature and Humidity in Brick Farmhouses.

| | | Minimum Value | Maximum Value | Average Value | Standard Deviation | Variance |
|---|---|---|---|---|---|---|
| B1 | Temperature (°C) | 5.276951 | 20.025177 | 12.42265321 | 3.144733858 | 9.889 |
| | Relative Humidity (%) | 23 | 80 | 35.46 | 5.570 | 31.021 |
| B2 | Temperature (°C) | 1.276798 | 7.167545 | 4.10184646 | 1.307188409 | 1.709 |
| | Relative Humidity (%) | 80 | 96 | 90.84 | 2.527 | 6.384 |
| B3 | Temperature (°C) | −1.219959 | 0.748074 | −0.29292286 | 0.374274319 | 0.140 |
| | Relative Humidity (%) | 53 | 68 | 64.17 | 3.014 | 9.082 |
| B4 | Temperature (°C) | −1.382851 | 0.446327 | −0.32932605 | 0.471483247 | 0.222 |
| | Relative Humidity (%) | 62 | 75 | 70.82 | 2.863 | 8.196 |
| D2 | Temperature (°C) | 3.036545 | 8.443962 | 4.96344065 | 1.341701328 | 1.800 |
| | Relative Humidity (%) | 74 | 78 | 76.17 | 1.079 | 1.163 |

The south-facing B1 exhibited a considerable temperature difference of approximately 15 °C with heating. Relative humidity changes were closely related to temperature changes and the humidity environment was better, within the design range [28]. In contrast, non-south-facing B2 with heating not only exhibited lower temperature, but also its temperature fluctuation was similar to the first floor south-facing D2 without heating. Notably, the average relative humidity during the monitoring period was 90.84%, which was a rather extreme indoor humidity environment. The high humidity environment can have an adverse effect on heating in winter, thus demonstrating the importance of orientation for brick houses with heating.

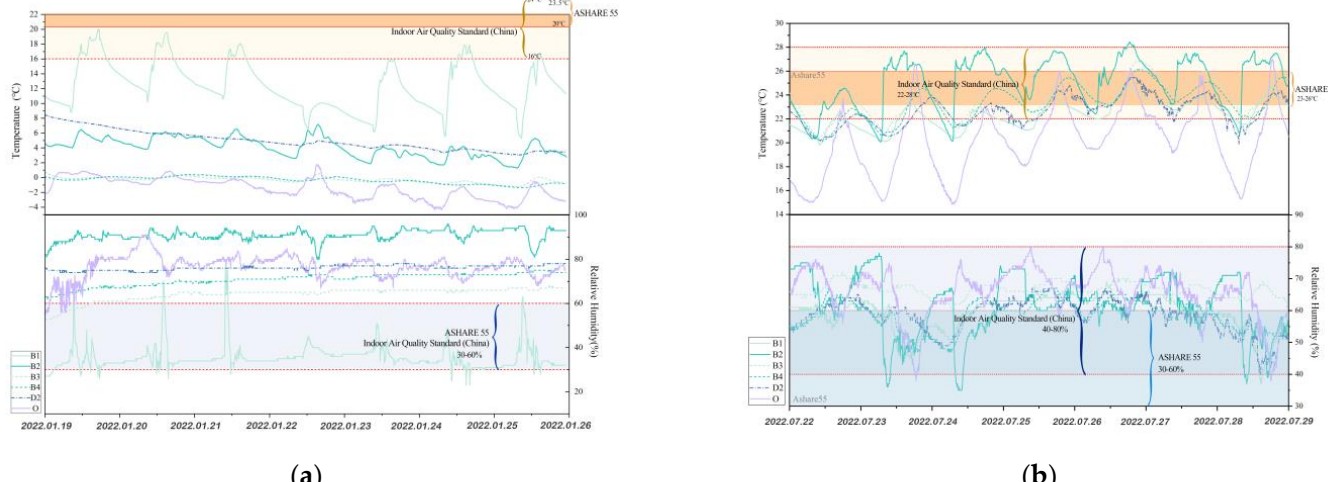

(**a**)                    (**b**)

**Figure 7.** Temperature and Relative humidity of brick houses in the monitoring week (RH: relative humidity). (**a**) Temperature and RH fluctuation in winter. (**b**) Temperature and RH fluctuation in summer.

The situation during summer is shown in Figure 7b. The difference in temperature due to orientation is not significant. The brick houses' temperature and relative humidity fluctuations are within the standard temperature range [28], and the humidity levels were influenced by the outdoor environment. It can be observed that the overall humidity levels in the brick houses are relatively closer to the recommended range set forth by ASHRAE.

*3.2. Comparison of Two-Story Brick Houses in the Village*

The study also monitored two-story brick houses in different areas of the village. All of the rooms were positioned in a southern direction D1 and D3 were heating with underfloor heating, but their temperatures still did not reach the design value [28]. The second stories of the buildings did not receive heating (see Figure 8) and the difference between them was not significant. But in terms of relative humidity, D2 exhibited higher levels in comparison to D4. It may be that the D4 building had been upgraded in terms of damp-proofing technology.

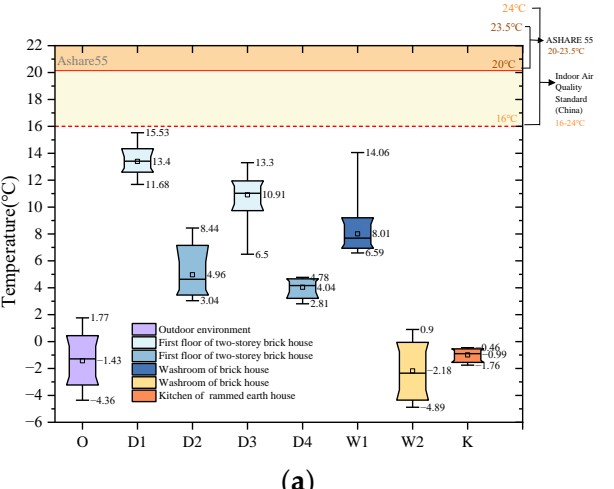

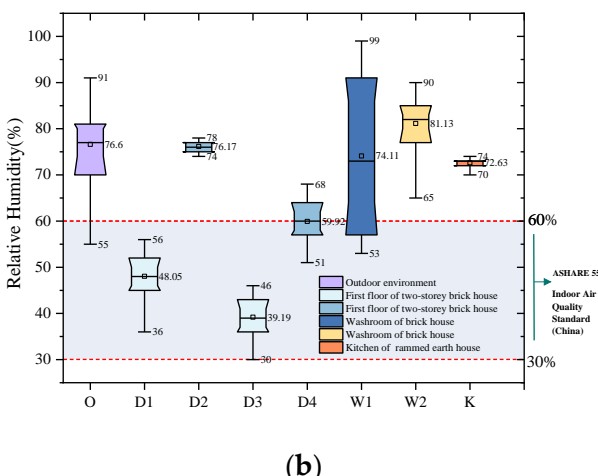

(**a**)                    (**b**)

**Figure 8.** Statistical box of plots of temperature and relative humidity of kitchen and washroom in winter. (**a**) Temperature in winter. (**b**) Relative humidity in winter.

Aimed at lavatories, W1 and W2 showed a large difference in temperature. W1 was a brick building on the outside, which showed high temperatures, due to its superior sealing. W2 was an earth building which was less well sealed, so that its indoor environment demonstrated no difference from the outdoor. This was also the case in K in the earth envelopes.

It was clear from the analysis that the residents of earth houses experienced several temperature steps during their indoor activities. According to previous research, higher temperature steps had a more serious impact on people's physical condition [30–32]. In order to avoid higher temperature levels, residents in the earth houses investigated in this research abandoned the function of rooms such as kitchen and chose to engage in other activities in rooms with heating. Residents in the brick houses were less affected by temperature steps and had a better overall functional utilization of their rooms.

## 4. Analysis of the Causes of High Humidity in Indoor Environments in Winter

### 4.1. Residential Monitoring Comparison

Under the heating condition, the difference in temperature of the south-facing brick house B1 was huge, significantly higher than for E1, E2, and B2. Furthermore, the relative humidity of B1 underwent considerable fluctuations. In the absence of heating, the distinction between the earth and brick houses in temperature was not significant, but the former exhibited a slightly higher relative humidity. The difference in temperature in the earth rooms experienced a gradual increase with heating, but it remained relatively stable. Conversely, the difference in the temperature of brick rooms increased more significantly (see Figure 9a). The relative humidity in both the brick and earth houses improved significantly with heating, although the design values were still not reached [28] (see Figure 9b).

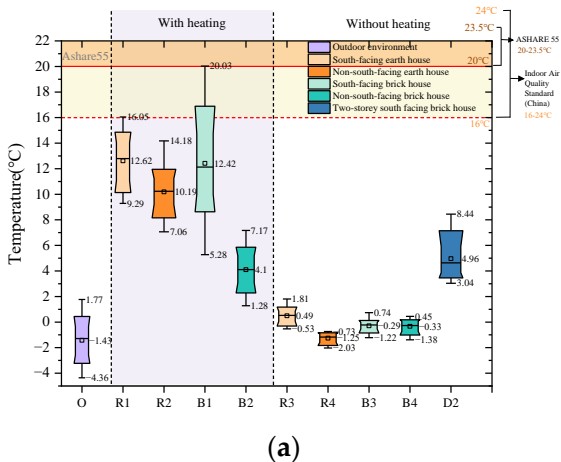
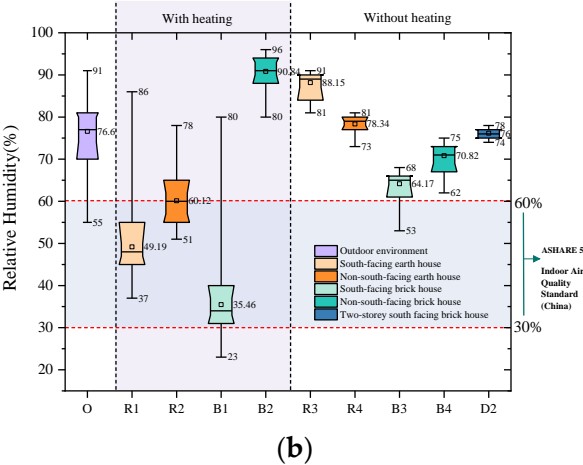

**Figure 9.** Statistical box of plots of temperature and relative humidity of all measured houses in first monitoring. (**a**) Temperature of first monitoring in winter. (**b**) Relative humidity of first monitoring in winter.

During the second monitoring period, measurements were taken focusing on the cases with heating. According to Figure 10, neither the temperature nor the relative humidity of earth houses were notably impacted by their orientation. And the average temperature in the south-facing rooms being slightly higher than in the non-south-facing rooms. Their temperatures were generally below the standard comfort temperature range and the relative humidity was high but relatively stable in terms of variation. If heating was continued, a more agreeable level of humidity could be attained. In the case of brick houses, they experienced more considerable fluctuations in both temperature and relative humidity compared to the non-south-facing rooms. The higher the temperature was, the greater the humidity fluctuation range would be (Figure 10).

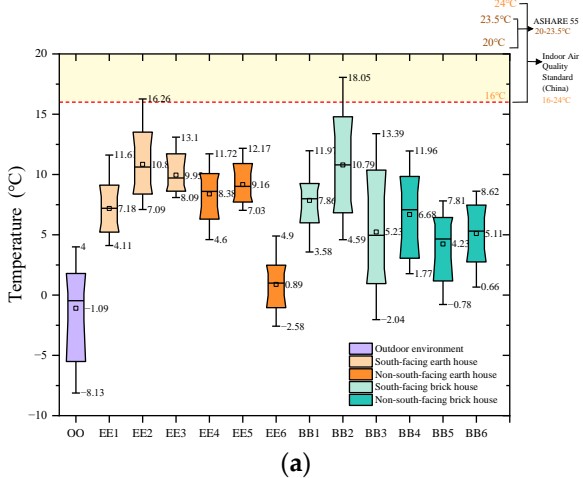

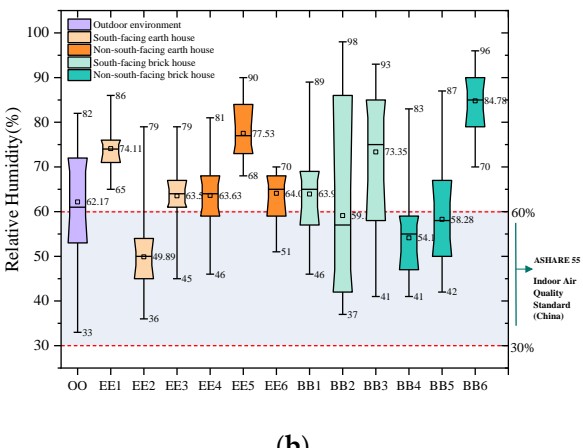

**Figure 10.** Statistical box of plots of temperature and relative humidity of all measured houses in second monitoring. (**a**) Temperature of second monitoring in winter. (**b**) Relative humidity of second monitoring in winter.

### 4.2. Analysis of the Correlation of the Indoor Thermal Environment in Earth Houses

The different types of building temperature and relative humidity were analyzed using SPSS 26. Regarding E4, the correlation between temperature and relative humidity yielded a $p$-value > 0.05, indicating an absence of discernible correlation. This may show that the non-south-facing earth houses without heating have stable relative humidity that did not vary with temperature. But there was an obvious negative correlation between the temperature and relative humidity in the south-facing earth houses. The reason of this may be due to the sunlight radiation which caused a partial increase in temperature which caused excess humidity to be evaporated (see Table 6).

**Table 6.** Temperature and Relative humidity Correlation Coefficients.

| Room No. | O | E1 | E2 | E3 | E4 | B1 | B2 | B3 | B4 | D2 |
|---|---|---|---|---|---|---|---|---|---|---|
| N | 2015 | 2015 | 2015 | 2015 | 2015 | 2015 | 2015 | 2015 | 2015 | 2015 |
| Pearson Correlation | −0.156 | 0.308 | −0.234 | −0.868 | 0.021 | −0.092 | −0.465 | −0.519 | −0.585 | −0.789 |
| Sig. (bobtail) | 0.000 | 0.000 | 0.000 | 0.000 | 0.351 | 0.000 | 0.000 | 0.000 | 0.000 | 0.000 |

This study elucidated the significance of the correlation coefficient between temperature and relative humidity. Positive values indicated that as temperature increased, relative humidity also increased, with higher values denoting a stronger association. Conversely, negative values indicated that as temperature increased, relative humidity decreased, and as temperature decreased, relative humidity increased, with lower values reflecting a stronger relationship. This value was therefore useful in helping to analyze the ability of the walls in terms of moisture buffering.

Pearson's correlation analysis using SPSS 26 of the daily mean temperature squared and daily temperature and relative humidity correlation coefficient values revealed that the data for the earth houses showed a significant correlation ($p < 0.005$). In contrast, no significant correlation was observed in the data for brick houses ($p = 0.981$). Through Origin 2021, a polynomial linear fit of the squared daily temperature values to the daily temperature and relative humidity correlation coefficients for the earth houses was achieved. It showed that, as the indoor temperature increased, the temperature and relative humidity correlation coefficients initially decreased and then increased. Two segments of data were then fitted linearly (see Figure 11), demonstrating that between −0.51 °C and 9.77 °C, the correlation showed a negative value, indicating that lower temperatures were associated with higher

relative humidity. In low-temperature, high-humidity environments, uncontrolled indoor temperature can worsen the situation. Moreover, when air temperatures was greater than 9.77 °C, a positive correlation between temperature and relative humidity emerged. It was a consequence of the full melting of the crystalline water in the envelope, which can cause moisture releasing effect [33,34].

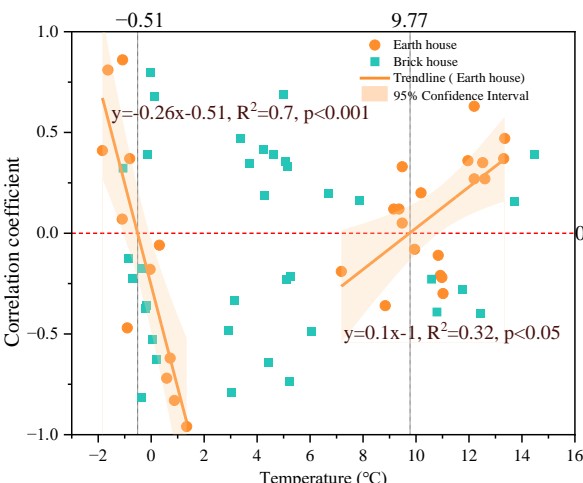

**Figure 11.** Relationship between temperature and the correlation coefficients of temperature and relative humidity.

## 5. Thermal Comfort of Residents

### 5.1. Thermal Sensation Vote (TSV) in Low-Temperature High-Humidity Indoor Environments

According to the American Society of Heating, Refrigerating and Air-Conditioning Engineers (ASHRAE), thermal comfort was classified into seven levels: very cold, cold, slightly cold, neutral, warm, hot, and very hot [35]. The results of this study were compared to those provided by the village's inhabitants, who rated the indoor environment during the winter months on a seven-point scale. Respondents were grouped by age: the younger group (18 to 44), the middle-aged group (45 to 60), and the elderly group (>60).

According to Figure 12, when considering the residential environment as a whole, the majority of the residents perceived their indoor environment as relatively cold, and the number of elderly who considered the indoor environment cold was the largest of the three. However, when analyzing the data based on the type of building, the highest percentage of the elderly group who perceived their environment as cold were residents of earth houses. Conversely, the highest proportion of young group perceived the environment to be cold in brick houses. This demonstrates that in brick houses, young residents are demanding a higher quality of the living environment.

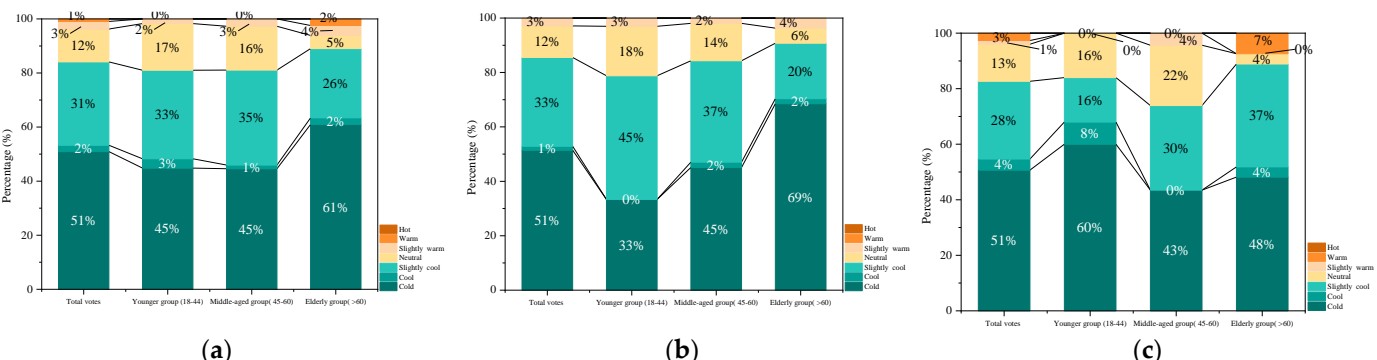

**Figure 12.** Winter indoor environment perception results. (**a**) Thermal sensation vote. (**b**) Earth house. (**c**) Brick house.

In summary, the overall results indicate that the local heating facilities were inadequate, leading to relatively harsh indoor conditions during the winter months.

The thermal sensation votes (TSV) from the 214 questionnaires were analyzed in relation to the corresponding ratings, with each data point representing the average of the TSV ratings at that temperature. The results then linearly regressed using Origin 2021 ($p < 0.000$). The slope of the resulting straight line represented the sensitivity of the population to temperature. The linear regression analysis of the thermal sensation votes of the residents in the brick houses was not significant ($p > 0.05$). In contrast, the residents in the earth houses showed a significant correlation, indicating that the residents in the latter were more notably influenced by environment (see Figure 13). Through the regression simulations, the residents of the earth house reported a neutral thermal sensation at temperatures approximately 22.5 °C.

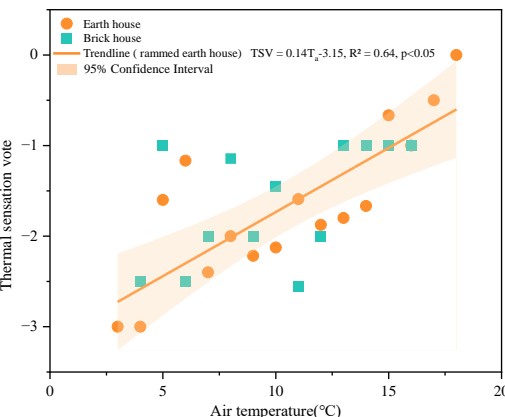

**Figure 13.** Air temperature and TSV in earth versus brick houses.

### 5.2. Indoor-Outdoor Temperature Steps and Thermal Comfort for Residents

In the 2023 supplementary questionnaire, respondents were presented with four choices to describe the impact of their living environment on their thermal comfort "no impact, slight impact, impact, and significant impact". Approximately 82% of the population chose that their living environment had a "slight impact" on their comfort. The remaining 18% who reported an "impact" exclusively resided in brick houses. For the indoor air quality, the options were: "−3 (very poor), −2 (poor), −1 (poor), 0 (fair), 1 (good), 2 (good), 3 (very good)". As all the people in the sample use Chinese Kang and coal stoves for heating, those who have lived in this environment for a long time generally rated the air quality under heating conditions as "fair and good". Those with a smoking habit generally scored lower.

As can be seen in Figure 14a, the mean disparity in TSV among the earth houses residents are generally close to the entire TSV score after moving from the indoor environment to the outdoor during the winter. The increase in temperature steps did not particularly affect the increase in the difference in scores or bring about differences in the scores of body parts. No significant associations were found when examining transient skin temperature changes and TSV scores, which is consistent with the findings of existing studies [36]. However, the TSV of the body parts of the residents in brick houses varies considerably. In particular, the difference in foot TSV was greatest for brick houses residents when the temperature changed, and there was a more significant correlation with temperature variation ($p < 0.05$). The graph presented after linear regression analysis by Origin 2021 is shown in Figure 14b. The difference in the evaluation of TSV in the foot increases with the temperature steps. And it has been shown that there is a strong correlation between foot temperature of healthy individuals and air temperature [37]. Supplementing heat to the feet in cold conditions can significantly improve the body's thermal comfort, so that heating from the feet is essential in the transition spaces of the region [38].

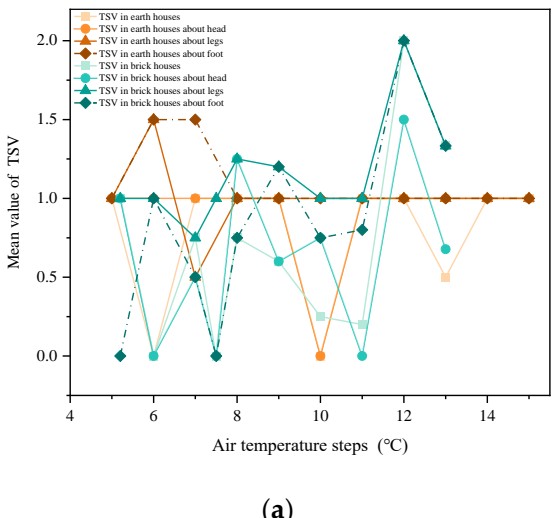

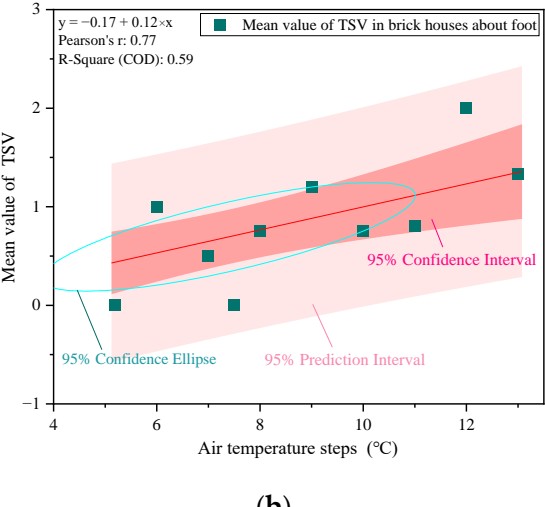

(**a**)

(**b**)

**Figure 14.** Relationship analysis between temperature steps and TSV. (**a**) Air temperature steps and mean TSV. (**b**) Relationship between temperature steps and TSV in mean foot position.

Simultaneously, the questionnaire results showed no significant correlation between changes in air temperature and changes in body temperature within 10 min. However, some residents experienced an increase in temperature within ten minutes of entering a colder environment. Residents of earth houses may have a correlation between their body temperature difference and air temperature difference, with $p = 0.074$ close to 0.05. It may be prudent to collect additional samples in subsequent stages to further explore this relationship. Accordingly, this study compared the mean indoor-outdoor TSV and thermal comfort vote (TCV) scores for the earth and brick houses with a small difference in the mean indoor-outdoor temperature steps and the mean body temperature difference, as shown in Table 7. It can be seen that almost all residents had lower TSV for their legs when indoors. Although their TSV scores of foot were rated relatively high, their TCV scores were in the lowest rating, which proves indicating a heightened need for temperature regulation in their foot. The legs also showed the least change in TSV and TCV ratings after going outside, which demonstrates that the legs are less sensitive than the foot and head. Previous studies have shown that the legs experience the least cooling when confronted with a drop in temperature [39]. It is worth noting that residents of earth houses show greater sensitivity to large temperature steps in their heads than residents of brick houses. This may be related to the relatively stable temperature and relative humidity of the living in the earth houses. It has been shown that after prolonged exposure to cold stimuli, the face took longer time to adapt to the cold compared to other locations, and after prolonged exposure to cold air, people's perception of cold changes decreased [40]. The sensitivity of the earth house aborigines is more significant as they have lived in a stable temperature environment for a long time and their facial adaptations are less well established. The difference between indoor and outdoor temperatures in this study was greater than 5 °C, and there was no significant change in the entire TSV ratings of the residents over a ten minute period, which is a significant difference from existing research [41]. This may be due to the fact that the local population has become acclimatized to such changes. However, simulations of the difference in TSV of foot and temperature step curves show that the conclusions of existing studies, in which temperature steps of 4 °C or less are sufficient for the body to regulate itself, still apply [41].

**Table 7.** Air temperature steps and Body temperature changes and mean score of TSV and TCV.

| House Type | Mean Value of Indoor Air Temperature (°C) | Mean Value of Indoor Body Temperature (°C) | Mean Value of TSV in Indoor Environments | | | | Mean Value of TCV in Indoor Environments | | |
|---|---|---|---|---|---|---|---|---|---|
| | | | Entire | Head | Legs | Foot | Head | Legs | Foot |
| Earth house (n = 19) | 15.368 | 36.463 | 0.000 | 0.211 | −0.053 | 0.211 | 1.000 | 0.789 | 0.737 |
| Brick house (n = 31) | 13.765 | 36.468 | 0.290 | 0.323 | −0.161 | 0.129 | 1.097 | 0.032 | -0.032 |
| House type | Mean value of outdoor air temperature (°C) | Mean value of outdoor body temperature (°C) | Mean value of TSV in outdoor environments | | | | Mean value of TCV in outdoor environments | | |
| | | | Entire | Head | Legs | Foot | Head | Legs | Foot |
| Earth house (n = 19) | 4.947 | 36.342 | −0.684 | −0.737 | −0.632 | −0.737 | −0.474 | −0.447 | −0.789 |
| Brick house (n = 31) | 4.484 | 36.374 | −0.290 | −0.290 | −0.452 | −0.774 | 0.419 | −0.032 | −0.516 |
| House type | Mean value of temperature steps (°C) | Mean value of body temperature changes (°C) | Difference between indoor and outdoor environmental mean TSV | | | | Difference between indoor and outdoor environmental mean TCV | | |
| | | | Entire | Head | Legs | Foot | Head | Legs | Foot |
| Earth house (n = 19) | 10.421 | 0.121 | 0.684 | 0.947 | 0.579 | 0.947 | 1.474 | 1.237 | 1.526 |
| Brick house (n = 31) | 9.281 | 0.094 | 0.581 | 0.613 | 0.290 | 0.903 | 0.677 | 0.065 | 0.484 |

According to the results of the 214 questionnaires, approximately 63% of the residents of earth houses suffered from chronic diseases, whereas this percentage was approximately 50% for the residents living in brick houses. Cold stimuli could significantly boost subjects' blood pressure [31,42]. The cardiovascular system has been shown to be sensitive to temperature steps [43]. In this study, the kitchens and washrooms of the earth houses were not heating and poorly sealed, resulting in a substantial temperature step compared to the heating rooms. This may confirm that temperature steps can be harmful to health.

Additionally, it is worth noting that despite the high incidence of rheumatism in the villages, the majority of residents in the follow-up survey were still dissatisfied with 55% relative humidity and expected the environment to continue to become wetter. This also reflected the fact that people's adaptation and need for relative humidity has been increased due to living in low-temperature and high-humidity environment for a long time.

## 6. Discussion and Limitations

There are numerous factors that contribute to the heightened indoor humidity levels in the region during the winter season. Primarily, the influence of the moist outdoor environment impacts the indoor atmosphere. The winter humidity in this area is relatively elevated due to the close proximity of villages to mountainous terrain and rivers. Consequently, the outdoor humidity during winter is greatly influenced by precipitation and hydrological conditions. The impact of outdoor hydrological environment on indoor buildings has been confirmed by multiple scholars' investigations [44,45]. Most scholars have discovered that water bodies possess the ability to reduce temperature and increase the surrounding humidity [46,47]. In China, the majority of rural areas are built near water bodies and surrounded by mountains and forests, which collectively contribute to the overall increase in humidity. Moreover, the local winter temperatures, combined with the absence of transitional spaces, prompt residents to seal their doors and windows for thermal insulation, resulting in inadequate ventilation. According to previous research findings, it has been shown that a lack of ventilation not only leads to increased indoor humidity, but also makes it difficult to decrease the concentration of carbon dioxide indoors. However, if the outdoor pollutants are excessively high, they can also impact the indoor environment [48]. Additionally, the moisture transmission capabilities of natural earthen materials diminish in lower temperatures, the wall has low thermal conductivity, and high heat capacity and significant thermal mass effect the key element enabling thermal stability [49]. Research has been conducted on the impact of temperature on humidity migration for similar building materials. It has been suggested that when the heat transfer

direction and moisture transfer direction are not aligned, it can lead to the evaporation of water vapor within the material [50]. This aligns quite well with the findings of the present investigation. Furthermore, this study has indicated that the addition of moisture barriers significantly reduces their moisture transmission performance [51]. Thus, the rational utilization of such materials in this region necessitates further exploration. The winter lifestyle habits of the local population also impact the humidity environment; inadequate ventilation also contributes to the retention of humidity [52]. Due to the lack of transitional spaces, kitchens are predominantly underutilized, prompting residents to conduct their daily activities within a single room. Consequently, the investigation of rational spatial functionality allocation was imperative.

A summary discussion of the limitations of this study is also presented here. The current research was unable to achieve absolute control over the dimensions and lighting conditions of all buildings, resulting in some slight variations. Additionally, there are inherent differences in construction techniques, which may have had a minor impact on the monitoring results. However, the monitoring method employed in this study still holds a certain level of universality and reliability in rural areas, making it suitable for studying different types of buildings in low-temperature and high-humidity environments. On the other hand, the regression simulation of the correlation coefficients between temperature and humidity in this study is currently in line with relevant findings, yet further research is needed for confirmation. In the investigation of thermal comfort, particular attention should be given to gender differences, as well as the differences in thermal comfort between individuals residing in low-temperature and high-humidity environments and those in more human-friendly living conditions. It is also worth considering additional influencing factors related to thermal comfort [53]. Future investigations could incorporate an inquiry into PMV-PPD (Predicted Mean Vote-Percentage of Dissatisfied) and undertake a comparative analysis. In addition, a well-maintained public hygiene environment is crucial for the development of rural areas. In future research, more emphasis should be placed on exploring the relationship between the architectural hygiene environment in rural areas and people's subjective perception and objective performance to develop strategies for improving the overall living conditions in these areas.

## 7. Conclusions

This research reveals a common winter indoor environment in rural China characterized by low temperature and high humidity. Insufficient attainment of the prescribed standards for indoor environmental conditions can contribute to an increase in building energy consumption. However, by effectively harnessing the inherent advantages of materials and enhancing the physical environment within the thermal context, the house can significantly reduce the energy consumption associated with the construction. Therefore, understanding the thermal performance mechanisms of different structural building materials is of vital importance as it provides crucial baseline values for environmental enhancement. In this study, monitoring of the indoor physical environment and a comprehensive questionnaire survey of an entire village population were carried out for 22 houses and 214 questionnaires in the southern region of Gansu, China. Furthermore, a detailed analysis of the environmental parameters and subjective questionnaires was conducted. The conclusions drawn from this study are as follows:

This study established a comparison of the thermal environments of earth and brick houses in areas characterized by low-temperature and high-humidity, finding certain differences between them. Orientation has little effect on the thermal environment of earth house compared to the brick houses, but a south-facing design remains a superior choice. Brick houses had substantial differences between their south-facing and non-south-facing rooms. In this study, the mean temperature of the south-facing rooms without heat was 4.96 °C, and of non-south-facing rooms, was −3.29 °C without heat for one week monitoring in winter.

The relative humidity of the earth houses remained high and was heated to stay in a comfortable relative humidity range, with a negative temperature/relative humidity correlation of approximately $-0.51\ ^\circ\text{C}$ to $9.77\ ^\circ\text{C}$. The indoor environment needs to be heated when the envelope loses its moisture buffering ability when the temperature is too low. Similarly, when the temperature is too high and the relative humidity rises, damp-proofing, and ventilation techniques need to be used.

The relative humidity inside the earth houses is more stable and a regression model can be built between TSV and temperature for long-term residents. And the residents in earth houses are more sensitive to changes in temperature. According to the results of the regression simulations, the residents of the earth house felt neutral at temperatures close to $22.5\ ^\circ\text{C}$.

The residents of brick houses have a highly variable environment and their TSV is not related to temperature alone. Their adaptability to temperature variations is higher. It is more appropriate to heat the transition space in brick rooms from bottom to top, and the temperature step between the transition space and the interior space should be controlled within $4\ ^\circ\text{C}$ in low-temperature high-humidity indoor environments.

This study explores the thermal comfort and building performance of specialized indoor and outdoor architectural environments. Future research could focus on the differences in the long-term health of residents compared to those residing in other types of environments; there may also be differences in psychological health issues. Additionally, further exploration of the relationship between indoor temperature and humidity could potentially serve as a means of measuring the performance of building materials during actual use.

**Author Contributions:** Conceptualization, J.L. and G.H.; Methodology, J.L. and X.W.; Investigation, J.L., X.W., S.K.W.C. and Q.Z.; Resources, S.K.W.C.; Writing—original draft, X.W.; Writing—review & editing, J.L., X.W. and Q.Z.; Visualization, J.L. and X.W.; Funding acquisition, J.L. All authors have read and agreed to the published version of the manuscript.

**Funding:** This research was funded by the Fundamental Research Funds for the Central Universities [Investigation, monitoring and renewal design of rural buildings under low-temperature and high-humidity environment] grant number 2022YJS125, the National Natural Science Foundation of China grant numbers 52078264 and 52078294 and the international cooperation projects of Scientific research project for low-carbon and energy-saving building Development Center II.

**Institutional Review Board Statement:** Not applicable.

**Informed Consent Statement:** Written informed consent has been obtained from the patients to publish this paper.

**Data Availability Statement:** The data presented in this study are available on request from the corresponding author. The data are not publicly available due to its inclusion of personal details of the researchers, which do not fall under the category of public data.

**Conflicts of Interest:** The authors declare no conflict of interest.

## Appendix A

Indoor and Outdoor Comfort Survey.

| Designation | Age | Gender | | Height (cm) | Weight (kg) |
|---|---|---|---|---|---|
| Attire description: | | | | | |
| Predominant room type for daily habitation: | | a. Earth house | b. Brick house | | |
| Heating method: | | | | | |
| Do you have any medical conditions? (Please specify) | | | | | |

| Designation | | Age | | Gender | | Height (cm) | | Weight (kg) |
|---|---|---|---|---|---|---|---|---|
| To what extent do you believe your physical ailments are influenced by your living environment? | No discernible effect 0 | Slightly impactful. 1 | Impactful 2 | Profoundly impactful 3 | | | | |
| Do you have a habitual inclination towards consuming alcoholic beverages? | a. Yes | b. No | Has the consumption of alcoholic beverages led to any deleterious health implications? | a. Yes | b. No | | | |
| Do you have a habitual inclination towards tobacco consumption? | a. Yes | b. No | Has act of tobacco inhalation led to any deleterious health implications? | a. Yes | b. No | | | |
| Please assess your state of bodily health | a. Optimal physical well-being | b. Albeit with occasional illnesses. | c. Ailments impede a harmonious existence. | | | | | |
| In the context of winter heating, please assess the indoor air quality? | Poor quality −3 | Deficient −2 | Slightly deficient. −1 | Neutral 0 | Slightly satisfactory. 1 | Satisfactory. 2 | Good quality. 3 | |
| Overall assessment of indoor environment: | Present indoor temperature: | | Current body temperature of the subject: | | | | | |
| | Thermal sensation vote | Cold −3 | Cool −2 | Slightly cool −1 | Neutral 0 | Slightly warm 1 | Warm 2 | Hot 3 |
| | Thermal comfort vote | Very uncomfortable −3 | Uncomfortable −2 | Just uncomfortable −1 | | Just comfortable 1 | Comfortable 2 | Very comfortable 3 |
| | Thermal acceptability vote | Very unacceptable −3 | Unacceptable −2 | Just unacceptable −1 | | Just acceptable 1 | Acceptable 2 | Very acceptable 3 |
| | Thermal sensation vote for head | Cold −3 | Cool −2 | Slightly cool −1 | Neutral 0 | Slightly warm 1 | Warm 2 | Hot 3 |
| | Thermal comfort vote for head | Very unacceptable −3 | Unacceptable −2 | Just unacceptable −1 | | Just acceptable 1 | Acceptable 2 | Very acceptable 3 |
| | Thermal sensation vote for leg | Cold −3 | Cool −2 | Slightly cool −1 | Neutral 0 | Slightly warm 1 | Warm 2 | Hot 3 |
| | Thermal comfort vote for leg | Very unacceptable −3 | Unacceptable −2 | Just unacceptable −1 | | Just acceptable 1 | Acceptable 2 | Very acceptable 3 |
| | Thermal sensation vote for foot | Cold −3 | Cool −2 | Slightly cool −1 | Neutral 0 | Slightly warm 1 | Warm 2 | Hot 3 |
| | Thermal comfort vote for foot | Very unacceptable −3 | Unacceptable −2 | Just unacceptable −1 | | Just acceptable 1 | Acceptable 2 | Very acceptable 3 |
| | Evaluate the overall level of humidity in the current environment | Very dry −3 | Dry −2 | Just dry −1 | Neutral 0 | Just humid 1 | Humid 2 | Very humid 3 |
| | Evaluate the level of air freshness in the environment | Very stuffy −3 | Stuffy −2 | Just stuffy −1 | Neutral 0 | Just fresh 1 | Fresh 2 | Very fresh 3 |

| Designation | Age | | Gender | | Height (cm) | | Weight (kg) |
|---|---|---|---|---|---|---|---|
| | Present indoor temperature: | | Current body temperature of the subject: | | | | |
| Overall assessment of outdoor environment: | Thermal sensation vote | Cold −3 | Cool −2 | Slightly cool −1 | Neutral 0 | Slightly warm 1 | Warm 2 | Hot 3 |
| | Thermal sensation vote for head | Cold −3 | Cool −2 | Slightly cool −1 | Neutral 0 | Slightly warm 1 | Warm 2 | Hot 3 |
| | Thermal comfort vote for head | Very unacceptable −3 | Unacceptable −2 | Just unacceptable −1 | | Just acceptable 1 | Acceptable 2 | Very acceptable 3 |
| | Thermal sensation vote for leg | Cold −3 | Cool −2 | Slightly cool −1 | Neutral 0 | Slightly warm 1 | Warm 2 | Hot 3 |
| | Thermal comfort vote for leg | Very unacceptable −3 | Unacceptable −2 | Just unacceptable −1 | | Just acceptable 1 | Acceptable 2 | Very acceptable 3 |
| | Thermal sensation vote for foot | Cold −3 | Cool −2 | Slightly cool −1 | Neutral 0 | Slightly warm 1 | Warm 2 | Hot 3 |
| | Thermal comfort vote for foot | Very unacceptable −3 | Unacceptable −2 | Just unacceptable −1 | | Just acceptable 1 | Acceptable 2 | Very acceptable 3 |

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
