# Peer review of "Thermal Comfort Comparison and Cause Analysis of Low-Temperature High-Humidity Indoor Environments of Rural Houses in Gansu Province, China"

_sustainability, doi:10.3390/su152316428_

Round 1

Reviewer 1 Report (Previous Reviewer 1)

Comments and Suggestions for Authors

This is an interesting indoor thermal comfort experimental and survey study focused on rural areas in china with significant population but low economic conditions.

Few minor comments for improvements:

General comments

Proof reading and language check required

Some of the figures are small and not clear, please revise

Please make sure to state the full term of abbreviations once introduced e.g. EIA, BMI

Please follow the journal’s standard reference style

Specific comments for each section

Abstract:

 ‘heating from feet’..please revise the term to be more scientifically sound

Introduction

‘Compared to urban areas, the development of energy efficiency in rural buildings remained comparatively.’ Please revise as this statement is not clearly understandable or complete.

‘Many houses were accustomed to using coal stoves and Chinese Kang[7][8]. ‘Check this statement was already mentioned two lines before, please avoid repetition.

‘At present, there was a wealth of research on the thermal environment of typical buildings in various climatic zones in  China101112. The current study used a small sample size of typical buildings. Existing studies lack  research on differences in the sensitivity of long-term rural inhabitants to temperature steps change  across different dwelling types.’… language problem, and the sequence of writing is not organized, seems that the sentence ‘The current study used a small sample size of typical buildings’ is cutting the reading flow. Please revise.

Datan village..spelling check

In February 2023, the team sampled 50 individualsin Datan… revise to be ‘In February 2023, the team sampled 50 individuals in Datan

Please show the House No. on the map or the layout

Please revise the structure and organization of the papers, seems that the authors have modified some sections e.g. in the result section 3.1.1. should be moved to section 2.1. where authors can include all background information relevant to the case study

Discussion

Please add more references to compare your findings against.

Comments on the Quality of English Language

moderate revision required

Author Response

Thank you for your comments concerning our manuscript entitled “Thermal comfort comparison and cause analysis of low-temperature high-humidity indoor environments of earth and brick houses in Gansu Province, China” (ID: sustainability-2666903). They are very helpful for research revision. After a careful understanding and study, our responds to the comments are as follows. Please refer to the attached document for the specific revisions.

Thank you for your hard work. We wish you a joyful life!

Kind regards,

Xijun Wu

Reviewer 2 Report (Previous Reviewer 3)

Comments and Suggestions for Authors

Unfortunately, only cosmetic improvements and broad conclusions for the future have been made. No hygienic and health requirements have been introduced in accordance with the standards.

Comments on the Quality of English Language At this stage, it is difficult to assess accurately

Author Response

Thank you for your comments concerning our manuscript entitled “Thermal comfort comparison and cause analysis of low-temperature high-humidity indoor environments of earth and brick houses in Gansu Province, China” (ID: sustainability-2666903). They are very helpful for research revision. After a careful understanding and study, our responds to the comments are as follows. Please refer to the attached document for the specific revisions.

Thank you for your hard work. We wish you a joyful life!

Kind regards,

Xijun Wu

Reviewer 3 Report (Previous Reviewer 4)

Comments and Suggestions for Authors

The article has some improvements.

Comments on the Quality of English Language

Problems are identified in the writing and citations, so it is recommended to use a style editor to improve the quality of the manuscript.

Author Response

Thank you for your comments concerning our manuscript entitled “Thermal comfort comparison and cause analysis of low-temperature high-humidity indoor environments of earth and brick houses in Gansu Province, China” (ID: sustainability-2666903). They are very helpful for research revision. After a careful understanding and study, our responds to the comments are as follows. Please refer to the attached document for the specific revisions.

Thank you for your hard work. We wish you a joyful life!

Kind regards,

Xijun Wu

Round 2

Reviewer 1 Report (Previous Reviewer 1)

Comments and Suggestions for Authors

Thank you for the authors. The manuscript now is now ready for publication. 

Comments on the Quality of English Language

Minor revision

Reviewer 3 Report (Previous Reviewer 4)

Comments and Suggestions for Authors

The manuscript may be accepted for publication

Comments on the Quality of English Language

The quality of English is acceptable for this review

This manuscript is a resubmission of an earlier submission. The following is a list of the peer review reports and author responses from that submission.

Round 1

Reviewer 1 Report

Comments and Suggestions for Authors

It is an interesting study regarding the thermal comfort comparison and cause analysis of low-temperature high-humidity indoor environments of earth and brick houses in Gansu Province, China

general comments

Proof reading and language check, For example double-story, wrong spelling

Please revise reference style according to the journal’s guidelines.

 specific comments for each section

Abstract: Please add more details about the research method e.g. the survey sample, and some percentage showing thermal comfort, satisfaction. Numbers shall increase the reliability of the research presented

In the method section: Please add floor plans to show the typical distribution of spaces in the houses investigated.

Are the houses composed of a single-story building?

Also a layout showing the houses and their location and orientation shall be much appreciated because it is not clear if all the houses have the same orientation or not which shall definitely affect the indoor thermal comfort conditions. Please clarify.

In the method section: It is not clear why the second questionnaire addresses the current living conditions of the local population and how this affect the indoor thermal comfort. Please clarify.

214 valid questionnaires were returned: but please clarify their percentages in terms of age, gender type, health state as all these parameters affect indoor thermal comfort.

Also it is not clear how many houses were investigated in total, how many of them were built in bricks and how many using earth construction. Only in the conclusion section, it is mentioned that they were 22 houses, and in table 4 it is mentioned the number of houses for each, but please mention it earlier in the method section so as not to confuse the reader.

Please add details about the wall thickness, openings..etc.

It is also important to mention that all the buildings investigated were passively ventilated, hence, no mean of mechanical system was used.

The authors may consider deleting these details as they may not be connected to the research study ‘As the local population is predominantly minority nationality and they observe Muslim rules about cleanliness. They wash much of their body or at least wash their face, mouth, nose, hands, and feet before each time they go worshipping.’, or justify how they are relevant to the study.

All the monitoring and surveys were carried in the winter season, but what about the summer season. It is also important to see the performance of these housing units in both summer and winter seasons. Please justify

Discussion

Please describe any constraints or limitations to the study. If the results obtained may change by changing of some parameters. Also comment on the replicability and reproducibility of the study. Compare your findings to previous studies to showcase the novelty.

 The conclusion section: please remove the numbering, it is better to keep it as paragraphs, pointing out the recommendations for future research at the end. 

Comments on the Quality of English Language

moderate revision required

Author Response

Dear reviewer:

Thank you for your amicable guidance.

The specific modifications and replies have been included in the Word document.

Wishing you a delightful life.

Sincerely,

Xijun Wu

Reviewer 2 Report

Comments and Suggestions for Authors

The research study investigates the indoor thermal conditions of earth and brick houses in the southern region of Gansu, China, focusing on areas with low-temperatures and high-humidity. The study measures environmental parameters like air temperature and relative humidity while also gathering residents' thermal sensations through surveys.

Below are comments that need to be addressed for the article to be positively considered for publication:

1) A clear novelty statement, which should express why this research is new or important, is missing.

2) Absence of a clearly stated research objective or aim.

3) Elaborate on the methodology and statistical analyses conducted.

4) Consider expanding the set of evaluated parameters for a more comprehensive understanding, such as PMV and PPD, to enrich the thermal comfort analysis.

5) The study concludes that orientation has little effect on earth houses, but doesn't sufficiently justify this with data.

6) Provide more comprehensive statistical analysis for the survey data to substantiate the claims made.

7) Lack of details regarding the questions included in the survey. It would be beneficial to attach the survey to the article.

8) Proofread and edit the paper for language and typographical errors to improve readability.

Author Response

(The authors gave the same response as above.)

Reviewer 3 Report

Comments and Suggestions for Authors

The presented tests of temperature, humidity and CO2 measurements do not refer to any standard of a residential house recognized in the world – area, height of rooms, layout. It should be emphasized here that the presented earth and brick houses do not have fixed insulation of the housing and the amount of air exchange inside the rooms, continuity of heating and the method of heat distribution, and this determines the reliability of research. In addition, the thermal perceptibility cited at work for different homes is not adequate to the hygienic and health conditions of living quarters in other countries. The article should be translated into an international standard and reference to other countries in the world.

Comments on the Quality of English Language

It needs to be improved.

Author Response

(The authors gave the same response as above.)

Reviewer 4 Report

Comments and Suggestions for Authors

a) To maintain coherence between the thematic line throughout the article, a more concise title can be presented as the current title is too long.

b) Instances of copied and pasted text from another source have been identified, such as lines 49 and 50, where the font remains unchanged, and line 154 exhibits the same issue.

c) In Figure 7, the caption should be included alongside the figure (refer to line 273).

d) There is a three-line break from line 360 until line 363, which needs to be corrected.

e) On line 392, separate the word from the number.

 f) The study presents important components, such as an analysis of the sensitivity to temperature changes in different areas of the body. It is suggested to focus more on the aspects discussed in the article instead of, for example, emphasizing the increase in energy consumption as a result of environmental improvements. Which may be true, but is not substantiated by your research.

Author Response

(The authors gave the same response as above.)

Round 2

Reviewer 1 Report

Comments and Suggestions for Authors

The authors have well-developed the manuscript based on the comments provided. 

Comments on the Quality of English Language

Minor revision required

Reviewer 2 Report

Comments and Suggestions for Authors

All recommendations have been appropriately addressed and incorporated into the manuscript. In my view, the article is in a state suitable for acceptance.

Reviewer 3 Report

Comments and Suggestions for Authors

The attached corrected article does not take into account the relevant comments and suggestions sent earlier. The content of the article shows that hygiene and health standards are not important.

Comments on the Quality of English Language

It needs to be improved